# The Psoriasis Therapeutic Potential of a Novel Short Laminin Peptide C16

**DOI:** 10.3390/ijms20133144

**Published:** 2019-06-27

**Authors:** Tsung-Chuan Ho, Shu-I Yeh, Show-Li Chen, Yeou-Ping Tsao

**Affiliations:** 1Department of Medical Research, Mackay Memorial Hospital, No. 45, Minsheng Rd., Tamsui District, New Taipei City 25160, Taiwan; 2Department of Ophthalmology, Mackay Memorial Hospital, No. 92, Sec. 2, Chung Shan North Road, Taipei 10449, Taiwan; 3Graduate Institute of Microbiology, College of Medicine, National Taiwan University, 7F, No. 1, Sec. 1, Jen-Ai Rd., Taipei 10617, Taiwan

**Keywords:** psoriasis, C16 peptide, α5β1 integrin, fibronectin, inflammation

## Abstract

Psoriasis is a chronic inflammatory skin disease characterized by excessive growth of keratinocytes and hyperkeratosis in the epidermis. An abnormality of the non-lesional epidermis at an early stage of psoriasis is involved in triggering inflammatory cell infiltration into the dermis. Integrin α5β1 acts as a receptor for fibronectin and has been found to be overexpressed in non-lesional psoriatic epidermis. To investigate whether α5β1 integrin has a potential as a drug target for psoriasis treatment, the α5β1 integrin-binding peptide, C16, was used to obstruct the HaCat keratinocyte cellular responses induced by fibronectin (Fn) in culture and psoriasis-like skin inflammation induced in mice by imiquimod (IMQ). The C16 exhibited antagonistic activity against α5β1 integrin in HaCat cells, with evidence of suppression of the Fn-mediated proliferative, cytoskeletal, and inflammatory responses. Topical treatment with C16 greatly reduced the IMQ-induced epidermal hyperplasia, infiltration of neutrophils/macrophages, and expression of pro-inflammatory mediators in mouse skin. The C16SP (C16-derived short peptide; DITYVRLKF) also exhibited antagonistic activity, suppressing α5β1 integrin activity in culture, and reducing IMQ-induced skin inflammation. Taken together, this study provides the first evidence that α5β1 integrin may be a potential drug target for psoriasis. The synthetic C16 peptide may serve as an agent for psoriasis therapy.

## 1. Introduction

Psoriasis is a chronic inflammatory skin disorder characterized by immune cell infiltration in the skin dermis, concomitant with hyperproliferation of epidermal keratinocytes, leading to a thickening of the epidermis and stratum corneum. The Interleukin-23 (IL-23)/IL-17/IL-22 cytokine axis, TNF-α, and IL-1β are important inflammatory mediators in the pathogenesis of psoriasis [1,2]. 

Integrins are a family of transmembrane receptors consisting of an alpha and a beta chain that interact with the extracellular matrix (ECM) to mediate cell behavior, including growth, differentiation, adhesion, and motility [3]. It has been reported that α5β1 integrin is overexpressed in the non-lesional psoriatic epidermis, compared with normal skin [4,5,6]. The overexpression of integrin α5β1 in keratinocytes of transgenic mice results not only in epidermal hyperproliferation, but also skin inflammation, providing a possible explanation for its pathogenic role [7]. In addition, integrin α5β1 is the primary receptor for fibronectin (Fn) [5,8]. Their interaction has been reported to induce proliferative signaling in endothelial cells and smooth muscle cells [9,10]. However, it is yet to be determined whether an α5β1 integrin antagonist is able to reduce the abnormal keratinocyte proliferation in psoriatic skin.

The synthetic C16 peptide (KAFDITYVRLKF) represents a functional domain in the laminin111 γ chain responsible for regulating angiogenesis [11,12]. Recently, C16 has been reported to have anti-inflammatory activity; systemic injection of C16 was shown to prevent leukocyte infiltration and alleviate detrimental inflammation in a rat model of acute allergic encephalomyelitis [13,14,15]. C16 is capable of binding to α5β1 integrin, shown by affinity chromatography, blocking the association of α5β1 integrin with Fn [12], but the influence of the C16 and α5β1 integrin association on keratinocytes, in response to Fn, remains unclear. 

Imiquimod (IMQ) is a toll-like receptor (TLR)7/8 agonist that has the ability to induce psoriasiform skin in humans and mice and to trigger a series of cytokines involved in the development of psoriasis, such as IL-23, IL-17, and IL-22 [2,16,17]. In this study, we investigated the effect of C16 on IMQ-induced psoriasis-like skin inflammation in mice. Our data show that C16 and a C16-derived short peptide (C16SP) could effectively alleviate the symptoms of psoriasis-like skin inflammation by inhibiting the hyperproliferation of keratinocytes, infiltration of neutrophils/macrophages, and generation of inflammatory cytokines. Our data indicate that the synthetic C16 and C16SP peptides are effective for the treatment of psoriasis in experimental animals. 

## 2. Results

### 2.1. C16 Peptide Disrupts the Fibronectin-Induced Cytoskeletal, Proliferative, and Inflammatory Responses in HaCat Cells 

HaCaT cells are immortalized human adult keratinocytes and are often used as a cellular model of psoriasis [18]. Moreover, expression of the integrins α5 and β1 in HaCat cells suggests that these play a role in maintaining keratinocyte proliferation in culture [18,19,20]. Our immunofluorescence staining study also showed the α5 integrin expressed in HaCat cells (Figure 1a). We first determined the effect of C16 on Fn-mediated focal adhesion (FA) formation. Serum-starved subconfluent HaCat cells were cultured on glass coverslips precoated with Fn. Immunofluorescence staining of FA components, including focal adhesion kinase (FAK) and F-actin, showed that nearly 86% of cells were expressed marked FA and formed fine stress fibers (Figure 1b,c; C16 solvent as control). In contrast, following C16 and C16SP treatment for 3 h, the FA-expressing cells were significantly reduced to levels of approximately 33% and 43%, respectively. Control peptide (NITYVALKF) had no effect on the α5β1 integrin/Fn-induced cytoskeletal responses. 

Next, the α5β1 integrin/Fn-induced cell proliferation was investigated. HaCat cells were cultured on a culture plate coated with Fn and incubated in low serum medium (2% FBS) containing 10 µM C16 or C16SP for 24 h. The numbers of cells were evaluated using a DNA-binding dye-based kit, showing that Fn-coating promoted HaCat cell proliferation compared to cells grown on an uncoated plate (Figure 1d; 124 ± 4% versus 100 ± 8%). The C16 and C16SP treatment substantially suppressed Fn-induced cell proliferation to levels of approximately 97% and 99%, respectively. Control peptide had no such effect.

Fn has been found in a soluble form in plasma and is abnormally expressed by dermal fibroblasts in the psoriatic non-lesional skin [5,6]. It has been reported that engagement of α5β1 integrin with Fn induces the NF-κB-dependent inflammatory program in endothelial cells [21]. We used TNF-α as an inflammatory marker to investigate whether C16 has the ability to suppress α5β1 integrin/Fn-mediated inflammation. HaCat cells were treated with both soluble Fn and C16 for 3 h and *TNF-α* gene expression was monitored by real-time qPCR. Soluble Fn induced *TNF-α* mRNA expression, approximately 21-fold greater than the untreated control cells (Figure 1e). However, cells treated with Fn in the presence of C16 and C16SP for 3 h led to 7.1-fold and 7.5-fold lower levels of *TNF-α* mRNA expression than cells treated with Fn/solvent. Taken together, C16 and C16SP can serve as an α5β1 integrin antagonist to impair Fn-mediating signaling in HaCat cells. 

### 2.2. Mitogenic Signaling Pathways Linking Integrin and Growth Factor Receptor in HaCat Cells are Blocked by C16

Psoriatic epidermis formed by the hyperproliferation of keratinocytes is one of major sources of inflammatory mediators in skin lesions [1,22]. We investigated the molecular mechanism of integrin and growth factor receptor signaling on HaCat cell proliferation to understand more how C16 provides a novel strategy for psoriasis therapy. Fn induces FAK autophosphorylation on the Tyr397 residue (p-FAK) that has been shown to be crucial for α5β1 integrin-mediated signaling cascades involved in cell adhesion, migration, and proliferation [23]. In addition, Tyr397 phosphorylation is a key event for subsequent full activation of FAK [24,25]. As expected, serum-starved HaCat cells treated with Fn underwent transient p-FAK induction at 5 min, assessed by western blot analysis (Figure 2a). As shown in Figure 1d, serum-starved HaCat cells, exposed to Fn in combination with 2% FBS (Fn/FBS), showed significant proliferation. Further, 2% FBS treatment increased the levels of p-FAK at 40~180 min compared to untreated cells (0 min). In particular, we observed that stimulation of cells with Fn/FBS caused a synergistic induction of the Tyr397 phosphorylation by ~2-fold, compared to 2% FBS, over the time period examined. Phosphoinositide 3-kinase (PI3K)/protein kinase B (Akt) has been reported to be one of the main signaling molecules positioned downstream of FAK [26]. Western blot analysis revealed that phosphorylation of Akt on Thr308 (p-Akt) was slightly increased upon FN stimulation at 5~40 min. HaCat cells exposed to 2% FBS had levels of p-Akt increased by ~2-fold at all time points (5~180 min) compared to untreated cells. Moreover, Fn/FBS substantially induced the levels of p-Akt by ~2-fold compared to 2% FBS at time points from 10 to 180 min. These results imply that the association of α5β1 integrin with Fn is linked with serum factors-mediated signaling to synergistically potentiate FAK/PI3K/Akt signaling in HaCat cells.

The involvement of FAK/PI3K/Akt signaling in HaCat cell proliferation was further evaluated using chemical inhibitors, including NVP-TAE226 and PF-573228 (FAK inhibitors) and LY294002 (a PI3K inhibitor). Inhibiting FAK or PI3K/Akt resulted in almost complete suppression of HaCat cell proliferation stimulated by Fn/FBS (Figure 2b). In addition, both NVP-TAE226 and PF-573228 dramatically abrogated the induction of p-FAK by Fn/FBS to the basal level (Figure 2c). FAK inhibitors also diminished the Akt phosphorylation induced by Fn/FBS. These results support the crucial role of FAK/PI3K/Akt signaling on Fn/FBS-mediated HaCat cell proliferation. Treatment with C16 or C16SP, but not control peptide, resulted in greatly decreased Fn/FBS-induced p-FAK and p-Akt, reaching a level similar to stimulation with 2% FBS alone. Collectively, the inhibitory effects of C16 and C16SP on Fn/FBS-induced mitogenic signaling in HaCat cells may be due to attenuation of the interaction of α5β1 integrin with Fn, leading to a blockade of integrin signaling, in synergy with serum-mediated signaling.

### 2.3. Enhanced Expression of Fibronectin and α5β1 Integrin in IMQ-Treated Skin

In the epidermis, basal keratinocytes use integrins to interact with the extracellular matrix (ECM) components at the dermal-epidermal junction (DEJ). It has been shown that integrin β1 is a predominant epidermal integrin expressed in basal cells of the epidermis and dermal fibroblasts in murine skin, whereas integrin α5 is weakly expressed [27]. In addition, it is well established that integrin β1 is critical for keratinocyte proliferation and skin integrity [27,28]. To determine the expression patterns of α5β1 integrin and Fn in murine psoriasis-like skin, the skin on the backs of mice was stimulated with IMQ for six consecutive days and their expression patterns were evaluated by immunofluorescence staining. As predicted, expression of integrin α5 and β1 was observed in normal epidermis, but a discontinuous staining often appeared in the epidermal basal layer adjacent to the DEJ (Figure 3a). Keratinocyte hyperproliferation was observed in the epidermis in IMQ-treated skin and integrin α5 and β1 were detected throughout the epidermal layers. The most notable feature was that the staining of integrin α5 was clearly stronger in the epidermal basal cell layer of IMQ-treated skin than in normal skin. In addition, we noted the accumulation of cell populations with high expression of α5β1 integrin in the dermis after IMQ stimulation.

Fn has been shown to be present in the upper dermis of normal murine skin, as judged by immunofluorescence [27]. As shown in Figure 3b, highly dense Fn staining in the dermis was a prominent feature in IMQ-treated skin, compared to normal skin. In addition, it was already known from in vitro studies that α5β1 integrin is responsible for polymerization of Fn [29]. Our immunofluorescence staining supporting the in vitro functional interaction between α5β1 integrin and Fn showed that an assembled Fn was aligned towards the integrin α5β1-enriched epidermal layer at the DEJ (Figure 3a,b; IMQ-treated skin). Taken together, our results indicate that IMQ stimulates the abnormal expression of α5β1 integrin and Fn in murine skin. The results imply that α5β1 integrin may interact with Fn at the DEJ after IMQ stimulation.

### 2.4. C16 and C16SP Reduce the Symptoms of Imiquimod (IMQ)-Induced Psoriasis

C16, C16SP, and DMSO vehicle were mixed with the IMQ cream to investigate their therapeutic effects on psoriasis. Subsequently, the mixed IMQ cream was applied topically to the shaved back skin and right ears of BALB/c mice, once per day for six consecutive days. On day 7, the dorsal skin of mice treated with IMQ/vehicle displayed apparent erythema and desquamation (Figure 4a). However, mice treated with C16SP showed a significant reduction of the erythema and desquamation in the back skin. Ear swelling is an inflammatory marker in this mouse model. IMQ/vehicle treatment increased the ear thickness by 1.9-fold, compared to the untreated control (Figure 4b). Treatment with C16 and C16SP reduced the ear thickening compared to IMQ/vehicle group (432 ± 21 and 437 ± 22 versus 592 ± 18 µm). However, control peptide did not suppress the IMQ-induced skin inflammation. Histological analysis of back skin by hematoxylin and eosin (H&E) staining also showed that the IMQ/vehicle group exhibited several important features of psoriasiform histology, including acanthosis (thickening of the epidermis), hyperkeratosis (thickening of the stratum corneum), and parakeratosis (retention of nuclei in the stratum corneum) (Figure 4c). In contrast, C16SP treatment resulted in a smoother epidermis than the IMQ/vehicle treatment, indicating the pathological phenotype was improved. Statistically, IMQ/vehicle significantly stimulated the epidermal thickening compared to normal skin (122 ± 10.5 versus 12.0 ± 0.8 µm), whereas C16 and C16SP led to a reduction in epidermal thickness (35.3 ± 4.1 and 39.2 ± 2.7 µm, respectively).

Hyperproliferation of keratinocytes is one of the main manifestations of psoriatic lesions. We monitored the IMQ-induced keratinocyte proliferation by BrdU labeling to investigate the acanthosis suppressed by the C16. Mice back skins were treated with IMQ for five days and then intraperitoneally injected with BrdU for another 24 h. Immunohistochemical analysis of the IMQ/vehicle group showed that the BrdU-positive cells were mostly distributed in the basal layer of the epidermis with dysregulated proliferation, in comparison with IMQ/C16SP (Figure 5a). Statistically, the numbers of IMQ-induced BrdU-positive basal cells were significantly reduced in mice treated with C16 and C16SP, compared with vehicle treatment (200× field: 8.5 ± 1.4 and 8.3 ± 1.4 versus 24.5 ± 2.0; Figure 5b). Collectively, these data suggest that C16 and C16SP are capable of improving the symptoms of psoriasis.

### 2.5. C16 and C16SP Suppress IMQ-Induced Inflammatory Responses in the Skin

The development of IMQ-induced psoriasis is dependent on several inflammatory cytokines. After IMQ/vehicle stimulation for four consecutive days, the levels of inflammatory cytokines were evaluated by real-time qPCR. The expression levels of the *TNF-α*, *IL-1β*, *IL-23*, *IL-17A*, *IL-22*, and *IL-6* genes were significantly increased in the skin stimulated by IMQ/vehicle, compared with the untreated control group (Figure 6; IMQ/vehicle set as 100%). Notably, these cytokine gene expression levels were markedly reduced in the IMQ/C16SP group, by a factor of 3~5-fold.

Infiltration of neutrophils and macrophages in psoriatic skin plays an important role in the promotion and persistence of chronic skin inflammation [2,17,30]. Here, we investigated whether C16 and C16SP could ameliorate the infiltration of neutrophils and macrophages induced by IMQ stimulation. Immunofluorescence staining of the lymphocyte antigen 6 complex locus G (Ly6g; a neutrophil marker) revealed that the numbers of Ly6g^+^ cells were greatly increased in the dermal layer of the skin after IMQ/vehicle stimulation for six consecutive days (Figure 7a). However, treatment with C16 or C16SP suppressed the neutrophil accumulation, compared to IMQ/vehicle treatment (Ly6g^+^ cells per 1000× field: 7.7 ± 1.6 and 7.9 ± 1.2 versus 18.6 ± 2.2; Figure 7b). In addition, immunofluorescent staining of the F4/80 (a macrophage marker) showed that C16 and C16SP effectively suppressed the macrophage infiltration, compared to IMQ/vehicle treatment (F4/80^+^ cells per 1000× field: 8.0 ± 2.2 and 9.7 ± 1.7 versus 25.7 ± 3.8).

Neutrophil chemoattractants CXCL1/CXCL2 (CXC chemokine ligands 1/2) are expressed abundantly in lesional psoriatic skin and are produced by keratinocytes upon stimulation by T helper type 17 (Th17) cytokines (such as IL-17 and IL-22) and TNF, contributing to pathogenic recruitment of neutrophils [2,31]. Th17 cytokines also induce psoriatic skin expressing macrophage chemokines, including CCL3 and CCL4 [2,32]. Furthermore, macrophages activated by inflammatory triggers also produce CXCL1 and CXCL2 [33]. To directly link the reduced infiltration of neutrophils and macrophages observed in the mice treated with both IMQ and C16, we investigated the expression of these chemokine genes by real-time qPCR assay. *CXCL1/2* and *CCL3/4* mRNA levels were significantly increased in the skin stimulated by IMQ, compared with the cream vehicle control group (Figure 7c). In contrast, expression of these chemokine genes was significantly reduced in the IMQ/C16SP group, by a factor of ~2.5-fold. Taken together, our results indicate that C16 and C16SP can ameliorate IMQ-induced skin inflammatory responses in mice, including infiltration of neutrophils and macrophages, as well as expression of a series of inflammatory genes.

## 3. Discussion

Integrin α5β1 expression has been reported to be maintained at a steady state in normal human adult epidermis [5,34]. Previous studies have shown that increased expression of α5β1 integrin in the epidermis and Fn enrichment at the DEJ are features of the non-lesional skin of psoriasis patients [4,5,6,18,34]. The role of α5β1 integrin on the pathogenesis of psoriasis has been highlighted in transgenic mice that overexpress α5β1 integrin in keratinocytes, with dramatic effects including epidermal hyperproliferation and inflammation [7]. We observed that the interaction of α5β1 integrin with Fn induced TNF-α expression in HaCat cells. This provides a molecular mechanism to support the pathogenic role of α5β1 integrin and Fn in skin inflammation. Furthermore, anti-TNF therapy by biological agents constitutes a first-line treatment of moderate-to-severe psoriasis [35]. In this study, C16 and C16SP showed marked efficacy in suppressing *TNF-α* gene expression in vitro and in vivo, showing an alternative pathway to control the TNF-α level in psoriatic skin. In the murine IMQ-induced, psoriasis-like skin inflammation model, we found that the amounts of α5β1 integrin and Fn proteins were dramatically increased at the DEJ, strikingly similar to their expression in human non-lesional psoriatic epidermis and psoriatic plaques. Therefore, this animal model may be appropriate for evaluating drug candidates targeting the α5β1 integrin. To the best of our knowledge, the current study is the first to show that an integrin antagonist has therapeutic potential for psoriasis.

It is well known that growth factors-associated matrigel or the ECM has the ability to regulate cell proliferation, in contrast with growth factors-reduced ECM [36]. In addition, growth factor-free Fn-coated slides are unable to induce HaCat cell proliferation [36]. Therefore, cellular behavior, such as proliferation, is regulated by the crosstalk between integrin signals and growth factor signals. FAK plays a critical role in the convergence of growth factor and integrin signaling pathways. For example, synergism in induction of autophosphorylation of FAK on Y397 has been observed in cells stimulated with Fn and angiotensin II [37], as well as Fn and platelet-derived growth factor (PDGF) [26,38]. Clustering of integrin by Fn induces FAK autophosphorylation and results in activation of several downstream pathways, including PI3K/Akt [26,39,40]. Also, Akt is a downstream target of growth factor receptors and can directly phosphorylate FAK at Ser695 and Thr700 that is essential for the recruitment of Src kinase to form the FAK-Src complex [39,41]. The FAK–Src complex facilitates FAK autophosphorylation induced by growth factors, subsequently, in synergy with integrin-mediated FAK signaling to control cell behavior [39,41,42]. Our result reveals that FAK autophosphorylation follows Akt phosphorylation induced by 2% FBS in HaCat cells, supporting the novel effect of Akt on FAK autophosphorylation. These molecular mechanisms may provide a possible explanation for our observation that 2% FBS synergizes functionally with Fn/integrin signaling in mediating HaCat cell proliferation. A previous study showed that C16 can bind to α5β1 integrin to block its association with Fn [12]. Our in vitro study supports the C16 effect with evidence that the connection of Fn/α5β1 integrin signaling and growth factor signaling on FAK/Akt is blocked by C16. The IMQ-induced psoriasis-like skin condition in mice is mediated by several crucial cytokines, of which IL-17A and IL-22 are known to activate PI3K/Akt signaling in epidermal keratinocytes to induce abnormal cell proliferation [43,44]. C16 effectively prevents IMQ-induced keratinocyte hyperproliferation in vivo. This result leads us to assume that the IL-23/IL-17/IL-22 axis connects with Fn/α5β1 integrin as a pathogenic mechanism in the development of psoriasis.

Here, we report that C16 and C16SP have beneficial pharmacological properties, such as blockage of neutrophil infiltration in psoriatic skin. Polymorphonuclear neutrophils (PMNs) are the most abundant circulating leukocytes of the innate immune system and PMN infiltration into the epidermis is a hallmark of psoriasis [1]. PMNs express several surface integrins, including α5β1, that are involved in their inflammatory infiltration [45]. However, whether C16 can interfere directly with PMN migration into the ECM awaits further investigation. In addition, interaction of α5β1 integrin with Fn has been shown to promote cell survival by activation of FAK-mediated signals [46]. This may be involved in the delayed apoptosis of PMNs during chronic inflammation [47]. The ability of C16 to block α5β1 integrin-mediated survival signaling in PMNs has yet to be investigated.

Integrin αvβ3 is a receptor for vitronectin and, reportedly, is not expressed in normal human keratinocytes [48]. Integrin αvβ3 is a predominant integrin expressed in inflammatory macrophages and participates in activation of the NF-κB-mediated inflammatory signaling [49]. Notably, inflammatory macrophages play an important role in the progression of psoriasis; they release IL-23 to drive and maintain the differentiation of Th17 [50]. C16 has been reported to be an αvβ3 integrin antagonist and to improve neuroinflammatory disease in experimental animals [13,14,15]. In our study, it still remains uncertain whether C16 acts against αvβ3 integrin to define the macrophage infiltration in skin stimulated by IMQ.

Traditional systemic therapies (glucocorticosteroids, methotrexate, cyclosporine, and retinoids) for psoriasis are often not able to meet the desired treatment goals, and high-dose and/or long-term use may be associated with toxicity, resulting in organ damage, such as nephrotoxicity, hypertension, bone marrow toxicity, and hepatotoxicity [51,52]. Several classes of biologicals have been developed against inflammatory cytokines, including TNF-α, IL-22, IL-23, and IL-17, although they are not effective in all individuals with psoriasis [51,53]. In addition, these biologicals cause some side effects, particularly the development of infectious diseases and increases in cancer risks [51,52,53,54]. Currently, patient satisfaction regarding psoriasis therapy remains low, and new drugs with better efficacy, a lower price, and fewer adverse effects are needed. Anti-integrin-based therapy provides an alternative strategy to resolve inflammatory diseases, such as Lifitegrast, a new FDA-approved therapy for dry eye disease. Lifitegrast is designed to mimic the intracellular adhesion molecule-1 (ICAM-1) for blocking the leukocyte function-associated antigen-1 (LFA1 or integrin αLβ2), resulting in reducing the T cells homing to the ocular surface [55]. The systemic administration of C16 has been shown to treat acute experimental allergic encephalomyelitis in rats and to have high degrees of safety and efficacy [13,14,15]. These reports imply that C16 may have a potential application for psoriasis therapy via systemic administration.

## 4. Materials and Methods

### 4.1. Materials

The peptides were synthesized, modified for stability by acetylation at the NH_2_ termini and amidation at the COOH termini, and characterized by mass spectrometry (>90% purity) at GenScript (Piscataway, NJ, USA). Dulbecco’s modified Eagle’s medium (DMEM), fetal bovine serum (FBS), antibiotic–antimicotic solutions, and trypsin were purchased from Invitrogen (Carlsbad, CA, USA). Fibronectin (F0895), 5-bromo-2’-deoxyuridine (BrdU), TRITC-phalloidin, Hoechst 33258 dye and all chemicals were from Sigma-Aldrich (St. Louis, MO, USA). Antibodies for BrdU (GTX42641) and fibronectin (GTX34727) were from GeneTex (Taipei, Taiwan). Phospho-FAK (Tyr397) antibody (#3283), anti-FAK antibody (#13009), phospho-Akt (Thr308) antibody (#13038), and anti-Akt (pan) antibody (#4691) were purchased from Cell Signaling Technology (Danvers, MA). Antibodies for integrin alpha 5 antibody (ab25251), integrin beta 1 (ab179471), Ly6g (PE/Cy7, ab25514), and F4/80 (ab6640) were from Abcam (Cambridge, MA, USA). FITC-donkey anti-rabbit IgG and FITC-donkey anti-mouse IgG were purchased from BioLegend (San Diego, CA, USA). NVP-TAE226, PF-573228, and LY-294002 were purchased from Selleckchem (Houston, TX, USA).

### 4.2. Cell Culture

The human keratinocyte cell line HaCaT was maintained in DMEM, supplemented with 10% (*v*/*v*) FBS and antibiotic–antimicotic solutions at 37 °C in a humidified 5% CO_2_ atmosphere. HaCat cells were also cultured on glass coverslips precoated with the Fn (10 μg/mL), according to the manufacturer’s protocol.

### 4.3. Cell Proliferation Assay

HaCat cells (1 × 10^3^/well) were seeded in 96-well cell culture plates (Costar; cat. no. 3599) precoated with the Fn (10 μg/mL; 50 μL per well) and incubated in 10% FBS culture medium for one day. HaCat cells cultured in uncoated wells served as a control. Subsequently, cells were maintained in serum-free DMEM for 3 h and then switched to stimulation medium, consisting of DMEM supplemented with 2% FBS and 10 μM peptide, for a further 24 h. Cell proliferation was evaluated by the Cell Proliferation Assay Kit (BioVision; Catalog # K307-1000, Milpitas, CA, USA), following the instruction manual. All assays were performed in triplicate, and the experiment was repeated 4 times.

### 4.4. Western Blot Analysis

Cell lysis, SDS–PAGE, and antibodies use for immunoblotting were as described in a previous study [56]. The band intensity in immunoblots was evaluated with a Model GS-700 imaging densitometer (Bio-Rad Laboratories, Hercules, CA, USA) and analyzed using Labworks 4.0 software.

### 4.5. Psoriasis Mouse Model

All animals were housed in an animal room under temperature control (24–25 °C) and a 12:12 h light-dark cycle. Standard laboratory chow and tap water were available ad libitum. Experimental procedures were approved by the Mackay Memorial Hospital Review Board (New Taipei City, Taiwan) (project code: MMH-A-S-104-37, date: 1 January 2016–31 December 2018) and were performed in compliance with national animal welfare regulations (Council of Agriculture, R.O.C).

BALB/c mice (8-week-old females) were shaved on the back and psoriasis was induced by topical application of with IMQ cream, as described previously [17]. Accordingly, each mouse received a daily topical dose of 62.5 mg of commercially available IMQ cream (5%; Aldara; 3M Pharmaceuticals) on the shaved back, and 15 mg of IMQ on the right ear, for 6 consecutive days. Control mice (normal group) were treated with a control cream. For treatment, 25 µM C16 or dimethyl sulfoxide (DMSO as peptide vehicle) was mixed with the IMQ cream. Histologic analysis of back skin specimens was performed just after the termination of IMQ application (on day 7). Ear thickness was measured in duplicate using a digital micrometer before challenge and after IMQ treatment for 6 consecutive days.

### 4.6. Histopathological Examination

Formalin-fixed samples from the middle of the back skin were embedded in paraffin, sectioned (5 µm), and stained with hematoxylin-eosin. Images were captured using a Nikon Eclipse 80i microscope (Nikon Corporation, Tokyo, Japan) equipped with a Leica DC 500 camera (Leica Microsystems, Wetzlar, Germany). At least six different sections were examined per skin sample. The epidermal thickness was quantified from the photographs using a computer-assisted image analyzer (Adobe Photoshop CS3 10.0).

### 4.7. In Vivo Detection of DNA Synthesis

To measure cell proliferation, BrdU was used after reconstitution in DMSO at (80 mM). Of this BrdU stock, 15 µL was mixed with 90 µL PBS and injected intraperitoneally into the mice at 24 h prior to euthanasia with CO_2_. Formalin-fixed, paraffin-embedded skin specimens were deparaffinized in xylene and rehydrated in a graded series of ethanol concentrations. Slides were treated with 1 N HCl at RT for 1 h before immunohistochemistry. The slides were blocked with 10% goat serum for 60 min and then incubated with primary antibody against BrdU (1:100 dilution) at 37 °C for 3 h. The slides were subsequently incubated with peroxidase-labeled goat immunoglobulin (1:500 dilution; Chemicon, Temecula, CA, USA) for 20 min and then incubated with chromogen substrate (3,3′-diaminobenzidine) for 2 min before counterstaining with hematoxylin.

### 4.8. Immunofluorescence

Deparaffinized tissue sections or 4% paraformaldehyde-fixed HaCat cells were blocked with 10% goat serum and 5% bovine serum albumin (BSA) in PBS containing 0.5% Triton X-100 (PBST) for 20 min. Staining was performed using primary antibodies against FAK (1:100 dilution), fibronectin (1:100 dilution), Ly6g (1:100 dilution), or F4/80 (1:100 dilution) at 37 °C for 3 h. For staining of integrin α5 and β1 (1:100 dilution), sections were stained at 4 °C for overnight. The slides were subsequently incubated with the appropriate fluorescent-labeled secondary antibodies (1:500 dilution) at 37 °C for 1 h and then counterstained with Hoechst 33258 for 6 min. The slide was also stained F-actin by TRITC-phalloidin (25 μg/mL) at 37 °C for 1 h. The slides were then rinsed with PBST three times, mounted with FluorSave™ reagent (Calbiochem, La Jolla, CA, USA), and viewed with an epifluorescence microscope (Zeiss Axioplan 2 imaging; Zeiss, Oberkochen, Germany) equipped with a charge-coupled device camera (Zeiss AxioCam HRm, Zeiss) and quantification was performed using Axiovert software (Zeiss AxioVision Release 4.8.2, Zeiss).

### 4.9. Quantitative Real-Time RT-PCR

The total RNA extraction, cDNA synthesis, and real-time PCR were performed as described previously [56]. Primers used in the experiment are listed in Table 1.

### 4.10. Statistics

The data were generated from three independent experiments. All numerical values are expressed as the mean ± SD. Comparisons of two groups were made using the Mann–Whitney test. *p* < 0.05 was considered significant.

## 5. Conclusion

In conclusion, our data suggest that C16 peptide might act as an α5β1 integrin antagonist to attenuate the early onset inflammatory responses in psoriasis-like skin disease. Further explorations of the expression of α5β1 and αvβ3 in the inflammatory progression of psoriasis and establishment of in vitro models for research into the effect of C16 on immune cells are important to further support the integrins as important targets for psoriasis therapy.

## Figures and Tables

**Figure 1 ijms-20-03144-f001:**
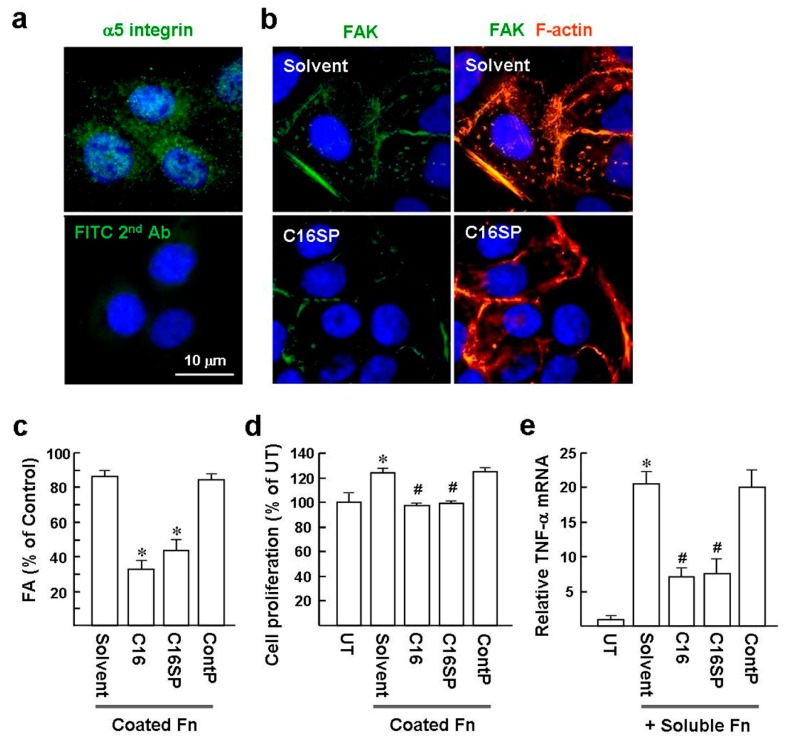
C16 peptide blocks α5β1 integrin-induced cellular responses in HaCat cells. (**a**) Immunofluorescence staining of α5 integrin. Representative images of three independent experiments. Nuclei were visualized with Hoechst 33258 stain. Original magnification, ×1000. NC: Immunofluorescence staining with FITC-conjugated secondary antibody alone. (**b**) Immunofluorescence staining. Subconfluent HaCat cells were plated on Fn-coated coverslips and incubated in low-serum medium (2% FBS) for 16 h prior to treat with 10 µM C16SP or solvent for a further 3 h. Representative images of three independent experiments show focal adhesion kinase (FAK) and stress fiber (TRITC-phalloidin staining). (**c**) The percentages of HaCat cells with marked FAK-positive focal adhesions (FA), per total cells, were calculated from 10 randomly selected microscopic fields in each treatment. Results are expressed as mean ± SD of three independent experiments. * *p* < 0.0001 versus solvent-treated cells. (**d**) Effect of the C16 on HaCat cell proliferation. Cell culture and treatment are described in the Methods. The proliferation index of each treatment was compared with the cells cultured on plates without Fn-coating (untreated; set as 100%). * *p* < 0.03 versus untreated (UT; Fn-uncoated and medium containing 2% FBS). ^#^
*p* < 0.004 versus solvent-treated cells. (**e**) 2 × 10^5^ HaCat cells were incubated in serum-free medium for 16 h, and then treated with Fn (5 µg/mL) and 10 µM peptide in fresh serum-free medium for another 3 h. Real-time qPCR analysis was conducted to determine the *TNF-α* mRNA levels. *Gapdh* was used as a loading control. Data are representative of three independent experiments. * *p* < 0.0003 versus untreated cells. * *p* < 0.001 versus solvent/Fn-treated cells.

**Figure 2 ijms-20-03144-f002:**
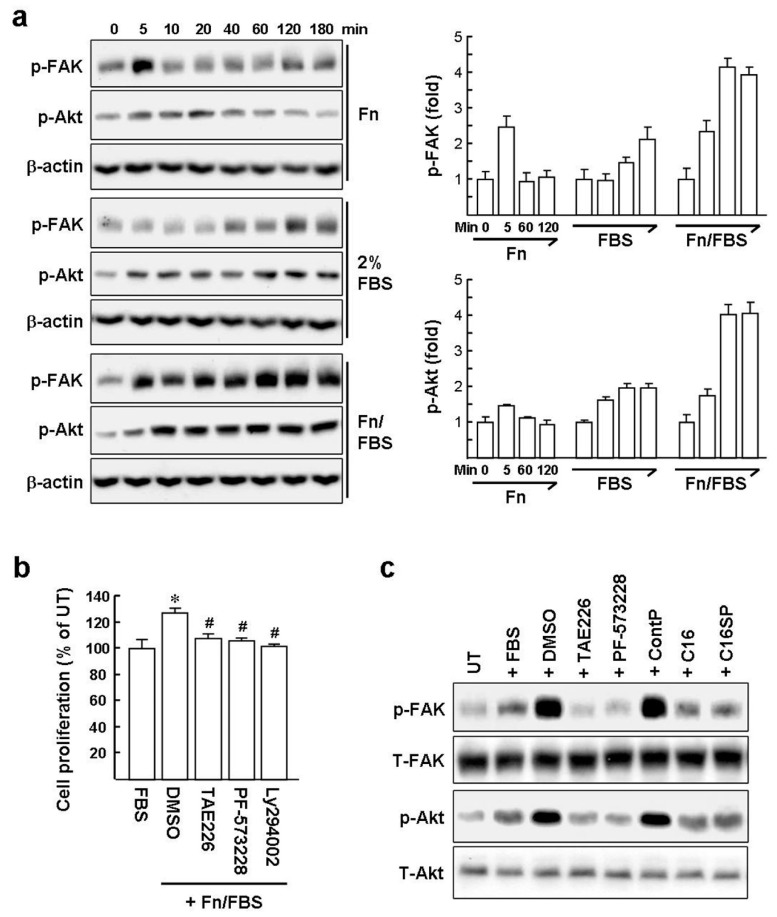
C16 and C16SP suppress Fn and serum-induced HaCat cell proliferation by blocking the FAK/PI3K/Akt signaling. (**a**) Fn and FBS synergistically induce phosphorylation of FAK at Y397 and Akt at T308 in HaCat cells in a time-dependent fashion. Cells were serum starved for 24 h and then treated with fresh medium containing 5 μg/mL of Fn or 2% FBS or both (Fn/FBS), for the time periods indicated. Representative immunoblots and densitometric analysis are from three independent experiments. β-actin served as a protein loading control. (**b**) Effect of the FAK inhibitors (NVP-TAE226 and PF-573228) and PI3K inhibitor (LY294002) on HaCat cell proliferation. Cells were serum starved for 24 h and then exposed to fresh medium containing 5 μM FAK or PI3K inhibitor for 30 min before addition of Fn/FBS for another 24 h. Cells treated with 2% FBS alone served as proliferative index 100%. Data represent three independent experiments, each performed in quadruplicate. * *p* < 0.005 versus FBS-treated cells. ^#^
*p* < 0.002 versus DMSO solvent-treated cells. (**c**) C16 and C16SP suppress Fn/FBS-induced phosphorylation of FAK and Akt. Serum-starved HaCat cells were treated with FAK inhibitor for 30 min or 10 μM C16 peptide for 60 min before Fn/FBS stimulation for 1 h. One representative immunoblot from three independent experiments is shown.

**Figure 3 ijms-20-03144-f003:**
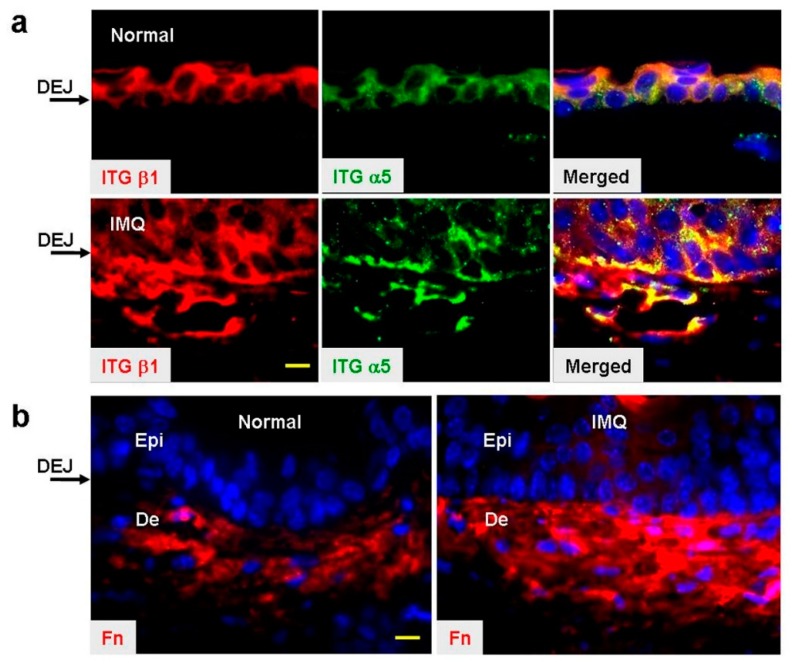
Immunofluorescence analysis of mouse dorsal skin six days after Imiquimod (IMQ) treatment. (**a**) Representative skin sections from mice treated with control cream (normal) or IMQ cream, and dual-immunofluorescence stained for integrin (ITG) α5 and β1. Nuclei were stained by Hoechst 33258 (blue). DEJ: Dermal–epidermal junction. (**b**) Representative images of fibronectin (Fn) staining of normal and IMQ-treated skins. Epi: Epidermis; De: Dermis. Scale bar: 10 µm. Results were evaluated from six sections of a skin specimen, with three mice in each group.

**Figure 4 ijms-20-03144-f004:**
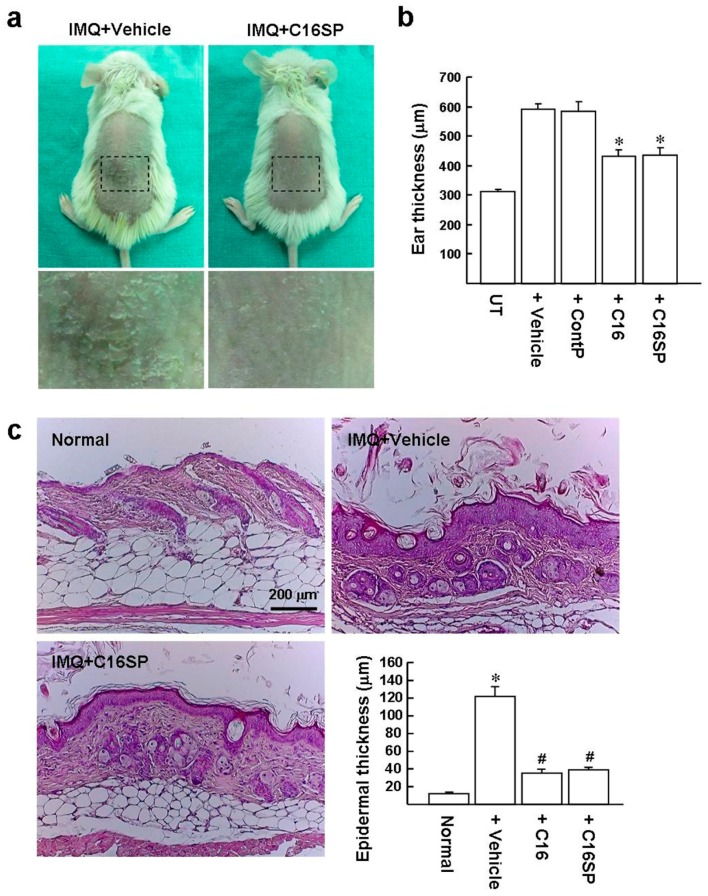
C16 and C16SP improve the symptoms of IMQ-induced psoriasis. BALB/c mice (eight-week-old) received a daily topical treatment with IMQ cream on the shaved back and right ear for six consecutive days. (**a**) Physical appearance of the back skin at the end of the study. Insert: Back skins photographed at higher magnification with a light-field camera. (**b**) Ear thickness measured by digital micrometer. * *p* < 0.001 versus IMQ/vehicle-treated group. (**c**) Histopathological examination of H&E stained dorsal skin. Representative images from three independent experiments with six mice per group are shown. The measurement of epidermal thickness is described in the Methods. Results are expressed as mean ± SD. * *p* < 0.0001 versus normal control group. ^#^
*p* < 0.0005 versus IMQ/vehicle treatment.

**Figure 5 ijms-20-03144-f005:**
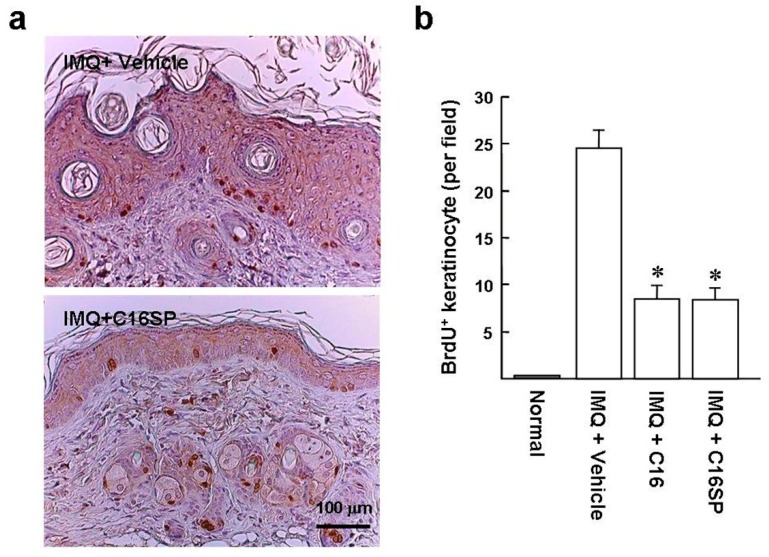
C16 and C16SP suppress IMQ-induced keratinocyte proliferation. Murine dorsal skins were treated with IMQ for five consecutive days and then intraperitoneally injected with BrdU for another 24 h, before being euthanized. (**a**) BrdU immunostaining (brown) and counterstaining with hematoxylin. Representative images are from six sections per mouse skin, with six mice per group. (**b**) Numbers of BrdU-positive cells in the basal layer of the epidermis. The digital image analysis of BrdU^+^ keratinocytes was performed blinded on an average of six randomly selected 200× magnification fields from each section, using a Nikon Eclipse 80i microscope equipped with a Leica DC 500 camera. * *p* < 0.0001 versus IMQ/vehicle-treated group.

**Figure 6 ijms-20-03144-f006:**
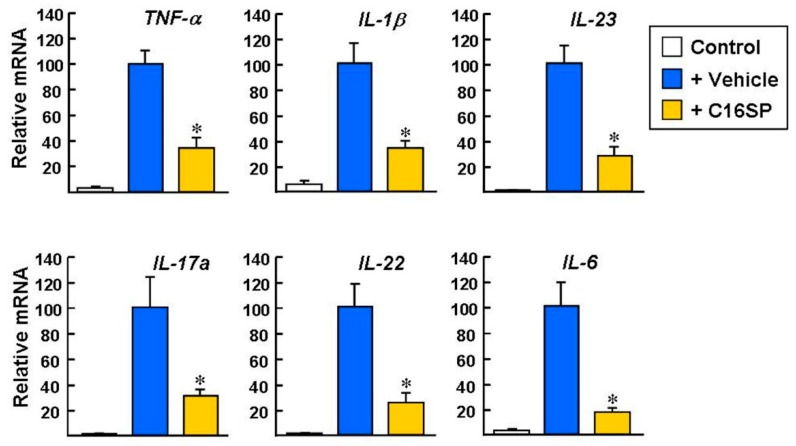
Real-time qPCR analysis of the expression levels of inflammatory cytokines induced by IMQ. Mice dorsal skins were treated with IMQ plus vehicle, C16, or C16SP for four consecutive days and then the dorsal skin samples were analyzed. Control indicates mice that received cream without IMQ. *Gapdh* was used as a loading control. Data are from three independent experiments. * *p* < 0.0001 versus IMQ/vehicle-treated group.

**Figure 7 ijms-20-03144-f007:**
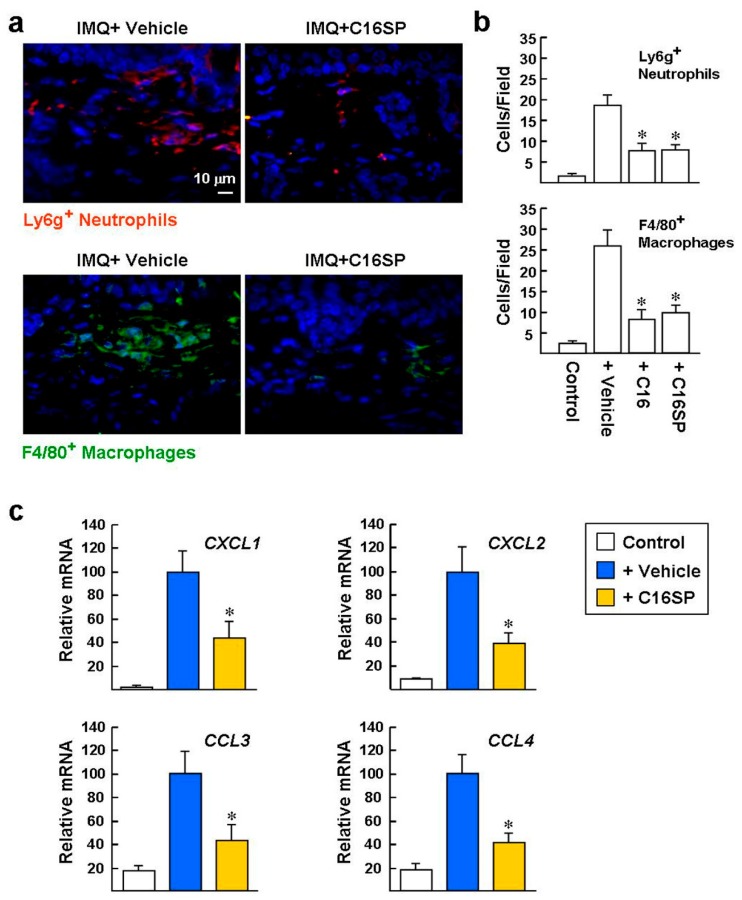
C16 and C16SP suppress IMQ-induced neutrophil and macrophage infiltration. (**a**,**b**) Mice dorsal skins were treated with IMQ cream plus vehicle, C16, or C16SP for six consecutive days. Immunofluorescence analysis of the levels of Ly6g^+^ neutrophils (*red*) and F4/80^+^ macrophages (*green*). Nuclei were visualized with Hoechst 33258 staining. Representative images are from six sections per mouse skin, with six mice per group. The digital image analyses of Ly6g^+^ and F4/80^+^ cells were performed blinded on an average of 6 randomly selected 1000× magnification fields from each section, using a Zeiss epifluorescence microscope and Zeiss software. Control indicates mice which received cream without IMQ. * *p* < 0.002 versus IMQ/vehicle-treated group. (**c**) Murine dorsal skins were treated with IMQ cream plus vehicle or C16SP for four consecutive days. Real-time qPCR analysis of the levels of chemokines. *Gapdh* was used as a loading control. Data are from three independent experiments. * *p* < 0.05 versus IMQ/vehicle-treated group.

**Table 1 ijms-20-03144-t001:** Primers used in the real-time qPCR.

Target Gene	Primer (Sense)	Primer (Antisense)
mIL-17A	5′-CTGCTGAGCCTGGCGGCTAC-3′	5′-CATTGCGGTGGAGAGTCCAGGG-3′
mIL-22	5′-CAGCTCCTGTCACATCAGCGGT-3′	5′-AGGTCCAGTTCCCCAATCGCCT-3′
mIL-23	5′-TCCTCCAGCCAGAGGATCACCC-3′	5′-AGAGTTGCTGCTCCGTGGGC-3′
mTNF-α	5′-GCCCACGTCGTAGCAAACCAC-3′	5′-GCAGGGGCTCTTGACGGCAG-3′
mIL-1β	5′-CCCTGCAGCTGGAGAGTGTGGA-3′	5′-TGTGCTCTGCTTGTGAGGTGCTG-3′
mIL-6	5′-CCTCTCTGCAAGAGACTTCCAT-3′	5′-AGTCTCCTCTCCGGACTTGT-3′3
mCXCL1	5′-ACCGAAGTCATAGCCACACT-3′	5′-CTGAACCAAGGGAGCTTCA-3′
mCXCL2	5′-TCCCTCAACGGAAGAACCAA-3′	5′-AGGCACATCAGGTACGATCC-3′
mCCL3	5′-ACATCATGAAGGTCTCCACCA-3′	5′-TCCATATGGCGCTGAGAAGA-3′
mCCL4	5′-AAGCCAGCTGTGGTATTCCT-3′	5′-CTCTCCTGAAGTGGCTCCTC-3′
mGAPDH	5′-GGGCTCTCTGCTCCTCCCTGT-3′	5′-CGGCCAAATCCGTTCACACCG-3′
hTNF-α	5′-AGGACCAGCTAAGAGGGAGA-3′	5′-TTCAGTGCTCATGGTGTCCT-3′
hGAPDH	5′-ACCCAGAAGACTGTGGATGG-3′	5′-TCAGCTCAGGGATGACCTTG-3′

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
