# Peer review of "The Psoriasis Therapeutic Potential of a Novel Short Laminin Peptide C16"

_ijms, 2019, doi:10.3390/ijms20133144_

Round 1

Reviewer 1 Report

The authors evaluate the potential role of α5β1integrin-binding  peptide, C16/C16-derived short peptide- in supressing psorisis inflammation in experimental models.  As previosly stated, overall impression on the article is that  it is soundly written, extensively documented and the research seems overall well designed and conducted. The conclusions are properly supported by the experimental data. All the suggested corrections have been properly addressed; therefore, I recommend the article for being published without further revisions.

Reviewer 2 Report

This is an intriguing study demonstrating the therapeutic potential of novel Short Laminin Peptide C16. The authors have responded to all the raised concerns and provided an adequate response. Newly performed experiments adds to the core message of the story and communicate the results in a mechanistic manner.

This manuscript is a resubmission of an earlier submission. The following is a list of the peer review reports and author responses from that submission.

Round 1

Reviewer 1 Report

This is an interesting study demonstrating the therapeutic potential of novel Short Laminin Peptide C16. The study clearly show the interaction between α5β1 integrin and fibronectin leading to cell proliferation in HaCat cells, which was suppressed by the laminin peptide C16 and C16SP. In the IMQ-induced psoriasis like skin inflammation model, both C16 and C16SP were able to suppress pathophysiological symptoms of the disease. Mechanistically, both C16 and C16SP alleviated psoriasis like skin inflammation by suppressing keratinocyte proliferation and infiltration of immune cells. The therapeutic potential of both C16 and C16SP is exciting, however there are following few points of concern which need to be addressed,

1) Interaction of α5β1 integrin and fibronectin should be shown in the mouse model as well.

2) What is the mechanism behind the anti-proliferative effects of C16 and C16SP in both in vitro and in vivo model (Figure 1d and Figure 3)? A discussion about this will add to the manuscript.

3) Treatment with C16 and C16SP reduced the number of infiltrating immune cells (neutrophils and macrophages) in the IMQ-induced psoriasis-like skin inflammation (Figure 4 a-b). In addition, C16SP treated mice demonstrated reduced expression of inflammatory cytokines (Figure 4 c). How about the expression profile of neutrophils and macrophages-specific chemokines in the skin of peptide treated mice? This would be a more direct link to the observed reduced infiltration of neutrophils and macrophages in the mice treated with IMQ and the laminin peptide.

4) In line 174-175, authors stated that “It releases IL-23 to attract Th17 cell”, is this statement is correct? IL-23 is a cytokine involved in Th17 cell differentiation. Are there some studies showing the chemotactic effects of IL-23? This should be cleared.

5) There are some grammatical errors which should be addressed and improved.

Reviewer 2 Report

The authors aimed to assess the potential role of α5β1 integrin-binding  peptide,  C16 and C16-derived short peptide in supressing psoriatic inflammation in both a cellular (HaCaT cells) and animal model. The article is well written, thoroughly documented and the research seems well conducted. The conclusions seem well supported by the experimental data.

However, several aspects are to be noted and corrected by the authors:

-The article may benefit from a graphical insert showing the interplay of molecules involved in psoriasis plaque development, emphasising the pathways that are being experimentally manipulated.

-please state the reason for chosing HaCaT cell line, stressing its advantages as a cellular model in line with the desired aim of the research.

-The article has numerous grammar and spelling flaws, that need to be corrected; some of them may not be just bothering (e.g. “that is sufficient to active the IL-23/IL-5317/IL-22 axis”, “C16 and C16SP peptides may be as a potent and safe agent”), but may also lead the reader into confusion regarding the conveyed message.

Some of these, although certainly not all of them, are listed below:

Line 15 –please rephrase “Integrinα5β1 is mainly as fibronectin receptor”

line 16 – please rephrase “... as a therapeutic target for the psoriasis treatment”

line 23  rephrase “In addition, the C16SP (C16-derived short peptide; DITYVRLKF) retained the C16 antagonistic activity was identified in culture and experimental animals”.

“It has been reported that α5β1integrin is remarkably overexpression” – please rephrase

Line 84 “The α5β1/Fn association has been found that promotes a NF-κB-dependent inflammatory program in endothelial cells”

Line 87 “HaCat cells were simultaneously treated with soluble Fn and C16 for 3 h, following the TNF-α gene expression”

“Notably, PMNs migrate on Fn is mediated by”

line 183 “Biologicals  have  been  developed to  against immunological factors, including TNF-α,”

line 187 “therapeutic challenge is still a need for new drugs with better  efficacy,  lower  price and  less  adverse  effects".

Line 188-189 “Anti-integrin  based  therapy  provides  an alternative strategy to resolve inflammatory diseases,such as Lifitegrast is a new FDA-approved therapy,  serves  as  a  T  cell  integrin”-please reformulate

line 192 “The report implies a possible of systemic treatment of psoriasis by C16. “-?

line 42 “Therefore, cell cycle promoted by the activated α5β1integrin may be a general mechanism” –the authors need to rephrase this as to convey a clarear message.

The Discussion section may be reorganized as to be clearer; in the actual form, it is a little too tortuous.

Moreover, in Section 4.9 –Was the data normality assessed beforehand? What were the statistical tests used in this respect – please specify. 

However, if data normality was not encountered, non-parametrical tests for two sample comparisons should have been employed insted of Student’s t test and this aspect needs to be stated adequately; in this case, the data should rather be presented as median, over mean.

Overall, the article is a useful contribution to the journal; i recommend it to be published, once the suggested corrections have been done.